# Current Management and Future Challenges in the Management of Severe Traumatic Brain Injury

**DOI:** 10.3390/medicina61040738

**Published:** 2025-04-17

**Authors:** Larissa Russo, Aasim Kazmi, Nasim Ahmed

**Affiliations:** 1Department of Surgery, Division of Trauma, Jersey Shore University Medical Center, Neptune, NJ 07753, USA; larissa.russo@hmhn.org; 2Department of Neurosurgery, Jersey Shore University Medical Center, Neptune, NJ 07753, USA; aasim.kazmi@hmhn.org; 3Hackensack Meridian School of Medicine, Nutley, NJ 07110, USA

**Keywords:** severe traumatic brain injury, current practice, challenges, outcomes

## Abstract

*Background and Objectives*: Severe Traumatic Brain Injury (TBI) is one of the devastating injuries occurring in all ages across the globe. Despite many advancements in the management of severe TBI, mortality and morbidities remain high. Evidence-based management in severe TBI has reduced mortality. The purpose of this review is to discuss the current management and present the future challenges in this patient cohort. *Materials and Methods*: A literature review was conducted to identify the current practice patterns and guidelines of severe TBI. We examined the literature regarding medical and surgical managements of the severe TBI. *Results*: Initial management of severe TBI includes stabilization of the primary injury and prevention of secondary insult to brain. Hemodynamic, intracranial pressure and cerebral perfusion pressure monitoring, antiseizure prophylaxis, hyperosmolar therapy, sedation, medical induced coma, and nutritional and ventilatory support are part of the medical management. Operative intervention includes craniotomy and decompressive craniectomy. Most of the current practices are recommended by the Brain Trauma Foundation (BTF). These guidelines are based on the existing literature, however, some of the recommendations by the BTF lack level one evidence. *Conclusions*: BTF guidelines provide recommendations in the management of severe TBI. High quality prospective randomized trials are needed to further explore the new modalities and interventions in the field of severe TBI.

## 1. Introduction

Globally, traumatic brain injury (TBI) remains a major cause of morbidity and mortality despite advances in care. Approximately 5.5 million people suffer from a severe TBI annually [1]. The rate of TBI has increased over time presumably due to increased motor vehicle accidents in developing countries and falls in the elderly in developed nations with aging populations. Outcomes in patients with moderate to severe TBI remain poor overall with an estimated 30% rate of full dependence or death by 6 months [2]. The total cost of managing TBI patients was reported to be more than USD 76 billion in 2010 in the USA alone [3].

Severe TBI patients are the most challenging patients to manage. The Brain Trauma Foundation (BTF) provides guidelines for managing TBI patients [4]. Last year, the American College of Surgeons published best practice guidelines for the management of TBI [5]. Most trauma centers in the US manage the severe TBI patient based on these guidelines. Implementation of BTF guidelines in the prehospital setting in the state of Arizona, USA, where severe TBI patients received early oxygen therapy and airway management to prevent hypoxemia and fluid bolus for prevention of hypotension, resulted in a reduction in mortality and morbidities [6]. The most recent update in managing severe TBI was published in 2020 [7] by Hawryluk et al. recommending updates in the existing BTF guidelines regarding operative management. Current management evolved over the last 25 years. However, there is no comprehensive review published that included recent advancements and outcomes in severe TBI management.

The aim of this review is to present current management practices and future challenges in severe TBI care. Our hypothesis was that the current management of severe TBI is based on the existing literature and recommendations from the BTF. This review describes current medical (hemodynamic, intracranial pressure and cerebral pressure monitoring, antiseizure prophylaxis, and other supportive measures) and surgical management including craniotomy and craniectomy in severe TBI.

## 2. Current Management

### 2.1. Surgical Management—Decompressive Craniectomy

Decompressive craniectomy (DC) is performed in patients with severe TBI to reduce intracranial pressure (ICP) by removing a portion of the skull. DC can be divided into two categories, primary or secondary, depending on the timing and reasoning for surgery [8]. Primary DC is performed at the time of removing a mass lesion while secondary DC is performed to treat refractory ICP [8]. The Brain Trauma Foundation’s Guidelines for the Management of Severe Traumatic Brain Injury provides four level-IIA recommendations for DC in severe TBI patients [7]. For late refractory ICP, the guidelines recommend that secondary DC be performed to decrease the risk of mortality and increase favorable outcomes. DC is recommended for refractory ICP to reduce ICP and time spent in the intensive care unit (ICU). A large frontotemporoparietal DC is suggested over a small frontotemporoparietal DC for improvement of survival and neurological recovery.

These recommendations are largely based on the results of two multicenter randomized controlled trials (RCTs)—Decompressive Craniectomy in Patients with Severe Traumatic Brain Injury (DECRA) and Trial of Decompressive Craniectomy for Traumatic Intracranial Hypertension (RESCUEicp). The DECRA trial randomly assigned 155 non-penetrating severe TBI patients (Glasgow Coma Scale (GCS) score 3–8 or Marshall Class III injury) between 15 and 59 years of age to receive bifrontotemporoparietal decompressive craniectomy or medical management within 72 h of injury [9]. At baseline, neither pupil was reactive in 27% of patients in the DC group and 12% of patients in the medical management group. During the 12 h before randomization, the median ICP was 20 mmHg in the two groups. The median time from injury to randomization was approximately 35 h and the median time from randomization to surgery in the decompressive craniectomy group was 2.3 h (interquartile range, 1.4 to 3.8). Patients in the DC group had lower ICP after randomization, and fewer ICU and ventilator days when compared to the medical management group. Patients in the DC group had worse scores on the 6-month Extended Glasgow Outcome Scale (GOS) but had similar 6-month mortality rates. The RESCUEicp study randomly assigned 408 TBI patients with raised ICP (ICP > 25 mmHg for 1–12 h) from 10 to 65 years of age to receive DC or to continue to receive medical care [10]. Baseline characteristics of the two groups were similar, except for history of drug or alcohol abuse. Pupillary abnormality was present in 29.2% of the DC group and 29.1% of the medical care group. A total of 53% of patients in the DC group and 50.0% of patients in the medical group had a GCS motor score of 1 or 2 at their first hospital while 47.0% of patients in the DC group and 50.0% of patients in the medical group had a GCS motor score 3–6. The DC was performed in this trial as third-tier management. Approximately 55–57% of patients were randomized in this study within 72 h of injury and remaining patients were randomized after 72 h of injury. The median time of surgery from randomization was 2.2 h [IQR; 1.3–5.1]. When compared to medical care, DC was associated with lower rates of mortality at 6 months but greater rates of vegetative state, lower severe disability, and upper severe disability.

### 2.2. Pharmaceutical Management

#### 2.2.1. Hyperosmolar Therapy

Hyperosmolar therapy is currently one of the standard treatments for elevated ICP and cerebral edema after severe TBI. There are two primary components of the mechanism of hyperosmolar therapy. One of the components is brain dehydration resulting from the osmotic pressure difference between brain cells and cerebral blood vessels [11]. The second component is blood viscosity reduction which improves microcirculatory flow of blood constituents and causes compensatory constriction of the pial arterioles, which lowers ICP and cerebral blood volume [11]. The most used hyperosmolar therapies in severe TBI patients are hypertonic saline and mannitol. A multicenter randomized clinical trial of adult moderate to severe TBI patients (GCS ≤ 12 and abnormal head computed tomography (CT) findings) found no difference in the 6-month Glasgow Outcome Scale-Extended (GOS-E) score and 6-month mortality when comparing patients receiving continuous infusion of 20% hypertonic saline in addition to standard care with patients receiving standard care only [12]. A prospective randomized controlled study of adult severe TBI patients receiving bolus infusion of either 20% mannitol or 3% hypertonic saline in an equiosmolar dose found no difference in 6-month GOS and 6-month mortality but a comparable decrease in ICP was observed between the groups [13]. A prospective randomized controlled study in adults with isolated severe TBI (GCS ≤ 8) and ICP > 20 mmHG for longer than 5 min compared three groups of patients receiving equiosmolar and isovolumetric doses of either 3% hypertonic saline, 20% mannitol, or a combination of 10% mannitol and 10% glycerol [14]. When osmotic load was similar, hypertonic saline, mannitol plus glycerol, and mannitol showed similar efficacy in lowering ICP. However, 3% hypertonic saline seemed to work better than the 10% mannitol plus 10% glycerol combination, and 20% mannitol. In a prospective RCT of children (≤16 years of age) with severe TBI (post-resuscitation pediatric GCS ≤ 8) and raised ICP receiving an equiosmolar dose of 3% hypertonic saline or 20% mannitol found that for the treatment of elevated ICP, hypertonic saline and mannitol had similar effectiveness [15]. Functional outcome was also similar between the groups. Overall, hyperosmolar therapy may reduce ICP in severe TBI patients, but the BTF guidelines state that there is not enough evidence on the effect of hyperosmolar therapy on outcomes to support a specific recommendation or a specific agent [4].

#### 2.2.2. Seizure Prophylaxis

Seizures are a common complication of severe TBI. Patients who develop seizures after severe TBI can experience increased ICP and metabolic emergencies which can lead to a more complicated hospital stay [16]. Phenytoin/fosphenytoin and levetiracetam are among the most commonly used anticonvulsants for the prevention of post traumatic seizures. Phenytoin is more cost-effective than levetiracetam, but phenytoin is associated with many side effects and drug interactions [16]. An RCT of moderate to severe TBI patients 5–50 years of age followed until the 7th day of injury found no difference in efficacy between phenytoin and levetiracetam in preventing early post-traumatic seizures [17]. A prospective randomized comparative trial of patients with severe TBI or subarachnoid hemorrhage (GCS 3–8 or GCS motor ≤ 5 and intracranial pathology on an abnormal CT scan on admission) receiving phenytoin or levetiracetam found no difference in early seizure incidence or seizure incidence at 6 months and no difference in mortality [18]. Patients in the levetiracetam group had lower 3-month Disability Rating Scale scores and higher 6-month GOS-E scores. As for side effects, the levetiracetam groups experienced less gastrointestinal problems and were less likely to have worsening neurological status. A prospective multicenter study of adult patients with blunt severe TBI (GCS ≤ 8 or GCS > 8 with subarachnoid hemorrhage, subdural hematoma, epidural hematoma, intracerebral hemorrhage or diffuse axonal injury) found no difference in early seizures, mortality, or adverse drug reactions between levetiracetam and phenytoin. However, the levetiracetam group had longer hospital lengths of stay [19]. Currently, the BTF recommends the use of phenytoin to reduce the risk of early post traumatic seizures if the benefit is thought to outweigh the risk of complications [4]. Phenytoin is not recommended as a prophylaxis for late post traumatic seizures. Despite the increase in use of levetiracetam in the severe TBI population, the BTF does not specifically recommend levetiracetam for prevention of early seizures due to insufficient evidence.

#### 2.2.3. Sedatives

In severe TBI patients, sedatives are commonly used for the facilitation of mechanical ventilation, to control agitation, and to lower ICP [20]. In the ICU, the most frequently used sedative agents are propofol and midazolam [20]. Prospective randomized studies comparing propofol and midazolam in severe TBI patients have reported no difference in cerebral metabolic profile [21], similar neurological injury marker serum concentrations and ICP [22], and no difference in GOS score at the time of ICU discharge [23]. When examining serum cortisol levels in propofol and midazolam patients, one study found that levels were reduced in both groups [23]. Two prospective randomized studies in severe trauma patients (including severe TBI patients), one comparing propofol, midazolam, and a combination of propofol and midazolam, and the other comparing propofol with midazolam, found no difference in ICP/cerebral perfusion pressure (CPP) or jugular venous oxygen saturation between groups in the TBI patients [24,25]. The BTF currently does not recommend propofol for improving mortality or 6-month outcomes but does recommend propofol to control ICP [4]. They advise caution when using high dose propofol as it can lead to significant morbidity. Alternative agents for sedation include ketamine and dexmedetomidine. Due to the potential for ketamine to increase ICP, it has been contraindicated in TBI patients in the past, but more recently, its use has increased because of its beneficial hemodynamics and respiratory properties [26]. In severe TBI patients in the ICU, dexmedetomidine can reduce agitation, prevent paroxysmal sympathetic hyperactivity, and decrease the incidence of delirium [27]. There is a risk for side effects including hemodynamic changes.

#### 2.2.4. Barbiturate Induced Coma

The BTF recommends administration of high-dose barbiturates for control of ICP that is refractory to maximum medical management and surgical treatment [4]. They advise that it is crucial to maintain hemodynamic stability prior to and during barbiturate therapy. Barbiturates can lower ICP by inhibiting brain metabolism, which lowers cerebral blood volume and metabolic demands [28]. Their ability to reduce ICP is believed to be a result of their ability to link cerebral blood flow to regional metabolic needs [28]. Barbiturates also lower blood pressure which can negatively impact CPP [28]. A prospective randomized cohort study comparing pentobarbital and thiopental in severe TBI patients with refractory intracranial hypertension that were treated with first-level measures from the BTF guidelines found that thiopental is more effective than pentobarbital for refractory intracranial hypertension [29]. The authors advise caution when interpreting results due to the small sample size, the unblinded nature of the study, different dosages in the groups, and differences in CT characteristics between groups. There was also no difference found between the groups in adverse events. A multicenter randomized trial in severe head injury patients with raised ICP found that adding pentobarbital to routine conventional treatment is useful for reducing ICP [30]. An RCT examining prophylactic pentobarbital in severe head injury patients found that the incidence, time, and treatment response of increased ICP did not differ between pentobarbital patients and the control group [31]. A larger percentage of patients in the pentobarbital group developed arterial hypotension when compared to the control group. Due to the results of this study, the authors did not recommend prophylactic pentobarbital coma for severe head injury. A prospective randomized study comparing pentobarbital and mannitol for intracranial hypertension following severe head injury did not find evidence that pentobarbital coma was superior to mannitol for ICP control or for increasing survival and suggested that pentobarbital coma could be harmful to patients without hematomas experiencing intracranial hypertension as a result of head injury [32].

#### 2.2.5. DVT Prophylaxis

TBI patients have an increased risk of developing deep vein thrombosis (DVT) following injury. The BTF states that mechanical prophylaxis can be used with low molecular weight or low dose unfractionated heparin for DVT prophylaxis [4]. Although, when combining mechanical prophylaxis with low molecular weight or low dose unfractionated heparin, there is a higher chance of intracranial hemorrhage expansion. If the brain injury is stable and the benefit is thought to exceed the risk of increase in intracranial hemorrhage, pharmacologic prophylaxis can be used in addition to compression stockings. There are currently no recommendations for the type, dose, or timing of pharmacologic agents for prevention of DVT.

### 2.3. Medical Management

#### 2.3.1. Prehospital Airway Management

Airway management of severe TBI patients is one of the key elements of preventing secondary insult. Hypoxemia can lead to the deterioration of injury and outcomes in severe TBI [33]. Therefore, it is extremely important to continue to diligently monitor the patient and provide intervention to prevent secondary insult with multimodal approach called “THE MANTLE” [34]. It is recommended to intubate the patient with severe TBI to maintain the airway and to provide constant flow of oxygen that the injured brain needs even during the transportation from scene to hospital [35].

#### 2.3.2. Ventilation

Mechanical ventilation is used in TBI patients for resuscitation, airway protection, tissue oxygen delivery, and regulation of cerebral vascular reactivity [36]. Despite this, mechanical ventilation increases the risk of respiratory complications. In the past, hyperventilation was proposed for the treatment of intracranial hypertension after TBI due to the ability of hypocapnia to decrease ICP [37]. However, more recent studies have found hypocapnia to be associated with poor functional neurologic outcomes [37]. Currently, the BTF does not recommend prolonged prophylactic hyperventilation with partial pressure of carbon dioxide in arterial blood (PaCO_2_) ≤ 25 mmHg [4].

#### 2.3.3. Cerebrospinal Fluid Drainage

Cerebrospinal fluid (CSF) drainage is a commonly used treatment for intracranial hypertension in severe TBI patients. CSF drainage lowers intracranial volume which leads to instant reduction in ICP and allows drainage of edema fluid into the ventricular system to reduce ICP over time [38]. CSF drainage can be performed using an external ventricular drain (EVD) or an external lumbar drain (ELD). According to the BTF, CSF drainage may be used in patients with initial GCS < 6 within the first 12 h of injury to decrease ICP [4]. Furthermore, continuous drainage of CSF with an EVD zeroed at midbrain may be more effective than intermittent drainage at decreasing ICP [4].

#### 2.3.4. Nutrition

Undernutrition in TBI patients can increase mortality risk, worsen neurologic outcomes, and lead to infectious complications [39]. Due to this, timely and efficient nutritional therapy is especially important. To reduce mortality after severe TBI, the BTF recommends patients be fed to achieve basal caloric replacement by at least the fifth day from injury and no later than the seventh day [4]. They also recommend the use of transgastric jejunal feeding to reduce the risk of ventilator-associated pneumonia.

### 2.4. Patient Monitoring

#### 2.4.1. ICP Monitoring and Thresholds

The BTF recommends ICP monitoring in severe TBI patients to reduce incidence of in-hospital mortality and mortality within 2 weeks of injury [4]. They recommend treatment of ICP > 22 mmHG due to the increase in the risk of mortality and state that ICP values combined with clinical and brain CT results may be used for treatment decisions. The BTF also provides recommendations for CPP. To reduce 2-week mortality risk, the BTF recommends the use of guideline-based recommendations for CPP monitoring. CPP between 60 and 70 mmHg is the target recommended by the BTF to increase survival and favorable outcomes. A multicenter RCT in severe TBI patients found no difference in 6-month mortality, ICU days, and serious adverse events when comparing patients receiving ICP monitoring with patients receiving imaging/clinical examination to guide treatment [40]. However, the imaging/clinical examination patients required more days of brain-specific treatments in the ICU, compared to the ICP monitoring group. A prospective randomized study comparing severe TBI patients receiving ICP monitoring with patients not receiving ICP monitoring found no difference in survival rate between groups [41]. Despite these studies finding no difference in mortality when comparing ICP monitoring to no ICP monitoring, there are observational studies that have reported ICP monitoring to be associated with decreased mortality [42,43,44,45,46,47,48].

#### 2.4.2. Blood Pressure Thresholds

Blood pressure maintenance plays a crucial role in the management of traumatic brain injury because it can help prevent secondary injury. A common secondary injury following TBI is hypotension. A recently published meta-analysis found a significant association between mortality and hypotension in moderate to severe TBI patients [49]. Currently, the BTF provides recommendations for systolic blood pressure (SBP) levels by age that may be used to lower mortality risk and enhance outcomes [4]. SBP ≥ 100 mmHg is recommended for patients 50–69 years of age while SBP ≥ 110 mmHG is recommended for patients 15–49 years of age or >70 years of age.

### 2.5. Post-TBI Rehabilitation Strategies

While some moderate and severe TBI patients have substantial long-term disabilities following their injury, many survivors have good functional outcomes after long-term rehabilitation [50]. During the acute care phase of TBI treatment, sustaining life and minimizing further brain damage are the primary goals [51]. After the acute care phase, the concentration is on optimizing daily functioning and returning to community living [51]. Some rehabilitation therapies for TBI survivors include psychotherapy, behavior therapy, vocational rehabilitation, occupational therapy, physical therapy, speech–language therapy, and cognitive rehabilitation therapy [51].

## 3. Future Challenges

### 3.1. Therapeutic Strategies

It is well established that the pathophysiology of TBI can be categorized into the primary injury as well as secondary injury. Primary injury occurs as a result of the direct impact of the external forces that lead to a TBI. This results in immediate damage to brain tissue and blood vessels. Secondary injury occurs as a cascade of physiological events that occur in response to the primary injury. While primary injury is irreversible, secondary injury can be mitigated or reversed through therapeutic strategies. Despite numerous pre-clinical trials showing promise, no single agent or combination of agents has been shown to be universally effective in improving outcomes.

#### 3.1.1. Tranexamic Acid

Tranexamic acid (TXA) is an antifibrinolytic synthetic derivative of the amino acid lysine that is used to reduce bleeding. A large randomized, placebo-controlled trial (CRASH-3) [52] evaluated the effects of TXA on patients with TBI. Expounding on the examiners’ previous study CRASH-2 [53] which showed that administration of TXA within 8 h of injury reduced mortality in general trauma patients, the authors sought whether it could also improve outcomes in TBI patients. In CRASH-3, 12,737 patients were randomly assigned to receive TXA or placebo within 3 h of injury. TXA was administered as a 1 g bolus followed by an infusion of 1 g over 8 h. When patients with GCS 3 or with bilateral unreactive pupils were excluded, administration of TXA was shown to reduce the risk of head injury-related death in mild to moderate TBI but not in severe TBI.

Another study by Rowell et al. [54] examined whether administration of TXA in the pre-hospital setting would improve neurological outcome in patients with moderate or severe TBI. In this multicenter, double-blinded, randomized clinical trial, 1063 patients were randomly assigned to receive one of three interventions: pre-hospital 1 g bolus and in-hospital 1 g infusion over 8 h, pre-hospital 2 g bolus and placebo infusion over 8 h, or pre- and in-hospital placebo. The primary outcome was the GOS-E at 6 months. The study showed no significant difference in a 6-month GOS-E, 28-day mortality, or progression of intracranial hemorrhage.

While TXA has not been proven to be effective in improving outcomes in severe TBI, there is a clear benefit in mild to moderate TBI. Further research is needed to determine its ultimate use in the management of severe TBI.

#### 3.1.2. Atorvastatin

The class of medications referred to as “statins” are well known for their lipid lowering effects. These HGM CoA reductase inhibitors also have antioxidant, anti-inflammatory, and anti-excitatory properties among other potential beneficial effects [55].

In a randomized double-blind placebo-controlled trial, Farzanegan et al. [56] examined the effects of atorvastatin on both brain contusion volume and functional outcomes in patients with moderate and severe TBI. Sixty-five patients were randomized to receive either 20 mg atorvastatin daily for 10 days or a placebo. Using CT imaging to measure contusion volumes on days 0, 3, and 7 post-injury, they found no significant difference in contusion volume but a significant improvement in functional outcome at 3 months post-injury.

In another study, atorvastatin was evaluated for its utility in the medical management of chronic subdural hematomas [57]. In this randomized, placebo-controlled double-blind phase II clinical trial, 196 patients received either atorvastatin 20 mg or placebo for 8 weeks. Patients who received atorvastatin showed both s significant reduction in hematoma volume and better neurological outcomes. It was postulated that the anti-inflammatory effects of statins were responsible for this benefit.

A number of other therapeutics have shown promise in pre-clinical studies including minocycline and candesartan. Minocycline is a tetracycline derivative antibiotic and candesartan is an angiotensin II receptor blocker. These medications and atorvastatin will be compared to placebo in an upcoming phase 2 multicenter, double-blind, placebo-controlled trial led by the TRACK-TBI initiative [58]. The results of this trial are eagerly anticipated.

#### 3.1.3. Amantadine

Amantadine is a dopamine receptor agonist that may promote and aid in the nervous system’s recovery in TBI patients [59]. A six-week randomized controlled clinical trial of TBI patients with GCS ≤ 9 measured the GCS score and FOUR score on the first, third, and seventh days after amantadine or the placebo was started and measured the Mini-Mental State Examination, GOS, Disability Rating Scale, and the Karnofsky Performance Scale 6 months after [59]. The study found a difference between the groups in the GCS score rising between the first and seventh days. The GCS score rising was better in the amantadine group. A six-week placebo controlled randomized clinical trial examining the effects of amantadine in severe TBI patients found that the change in the Disability Rating Score from baseline was better in the amantadine group [60]. However, there was no difference between the amantadine group and placebo group in GOS score. Severe TBI patients who were in a minimally conscious or vegetative state and were receiving inpatient rehabilitation 4 to 16 weeks post TBI were assigned to 4 weeks of amantadine or placebo in a placebo-controlled trial [61]. Disability Rating Scale scores showed faster recovery during the 4 weeks of treatment in the amantadine group. Two weeks after treatment, rate of improvement in the amantadine patients decreased and was slower than the rate in the placebo patients. Between baseline and week 6, overall improvement in the two groups’ Disability Rating Scale scores were similar. A randomized clinical trial comparing amantadine and a placebo in TBI patients who were intubated, admitted to the ICU, had a diffuse axonal injury, and a GCS ≤ 8 found no difference in admission or discharge GCS, GOS, duration of mechanical ventilation, hospitalization, and mortality between the groups [62]. A randomized controlled trial comparing amantadine, zolpidem, and a placebo for the first 8 days post-injury in severe TBI patients with GCS ≤ 8 found that amantadine increased the rate of GCS and GOS when compared to zolpidem and the placebo; however, there was no significant difference [63].

#### 3.1.4. Cerebrolysin

Cerebrolysin is a mixture of peptides and amino acid extracted from the porcine brain. As part of early therapy for moderate and severe acute TBI, cerebrolysin has been proposed to have positive effects on cognitive function, neuro-motor recovery, and neurobehavioral functioning [64]. CAPTAIN I and CAPTAIN II, randomized, placebo-controlled trials evaluating the safety and efficacy of cerebrolysin in moderate to severe TBI patients found that cerebrolysin had a beneficial effect on overall outcomes after TBI [64,65].

### 3.2. Biomarkers

Serum protein biomarkers have been identified which can rule out the need for CT imaging in mild TBI. In the ALERT-TBI prospective multicenter observational trial, it was shown that a test measuring ubiquitin carboxyl-terminal hydrolase L1 (UCH-L1) and glial fibrillary acidic protein (GFAP) had high sensitivity and negative predictive value (NPV) and could be used to rule out the need for obtaining a CT scan in mild TBI [66]. This led to the development of a point-of-care test that can provide values of GFAP and UCH-L1 via a portable handheld device within minutes.

In minimal to mild TBI, testing for serum biomarkers can rule out the need for CT imaging. The clinical utility of serum biomarkers in severe TBI, however, remains to be seen. The CENTER-TBI investigators showed that higher serum biomarker levels were associated with worse outcomes. It was shown that all six tested serum biomarkers (UCH-L1, S100B, GFAP, t-tau, neuron specific enolase, and neurofilament protein-light) improved the prognostic value for functional outcome measured by GOS-E at 6 months post-injury, with UCH-L1 having the greatest incremental prognostic value [67].

### 3.3. Advanced Neuromonitoring and Autoregulation

Advanced neuroimaging of magnetic resonance imaging (MRI), particularly diffusion tensor imaging, and volumetric analysis can identify injury that is not visible on regular CT scan or MRI. Functional MRI and MRI spectrometry are still in research arena that may progress to clinical application. The use of a protocol-based approach to manage traumatic brain injury has been shown to improve clinical outcomes. It is well accepted that a target ICP less than 22 mmHg and a cerebral perfusion pressure (CPP) 60–70 mmHg are established treatment goals for patients with severe TBI. However, due to variability in outcomes even when these guidelines are followed, it has become apparent that a more personalized approach within these guidelines may be necessary. This precision medicine approach is an innovative way to help improve patient outcomes. Incorporation of additional data into the monitoring algorithm such as brain tissue oxygenation and cerebral autoregulation may provide a path forward.

#### Cerebral Autoregulation

Cerebral autoregulation is the physiologic process of maintaining adequate blood flow to the brain despite changes in systolic blood pressure. Cerebral autoregulatory processes can be affected by severe TBI. The normal response to increased blood pressure is vasoconstriction to maintain cerebral blood flow; this, in turn, reduces arterial blood volume and results in reduced ICP. The autoregulatory response can be assessed by the Pulse Reactivity Index (PRx). If the PRx is above 0.25, where increases in mean arterial pressure (MAP) lead to increased ICP, autoregulatory means are compromised and this is associated with poorer outcomes. Negative PRx values indicate ICP does not rise with elevated MAP and are associated with improved outcomes [68].

The Seattle International Severe Traumatic Brain Injury Consensus Conference (SIBICC) developed consensus-based algorithms to aid in the management of ICP in patients with severe TBI. Due to a lack of recent evidence-based treatment algorithms, a consensus working group was convened in order to develop a tier-based approach to managing ICP. A three-tier system was recommended. As part of Tier 2, a MAP Challenge may be used to assess cerebral autoregulation. Increasing CPP can result in decreased ICP if cerebral autoregulation is intact. During the challenge, administration of a vasopressor or inotropic agent is titrated to increase MAP by 10 mmHg for up to 20 min and MAP, ICP, CPP, and brain tissue oxygenation (Pbt02) (if available) are monitored closely before, during, and after the test. If autoregulation is not intact, raising MAP will concomitantly raise ICP. These data can then be used to assess whether the benefit of lowering ICP justifies the risk of pharmacologically raising MAP [69].

The BTF guideline range for CPP between 60 and 70 mmHg is well regarded. Within that range, however, there may be an optimal CPP target (CPPopt) which is dependent on the autoregulatory status of the patient. In the COGiTATE feasibility trial [70], investigators analyzed whether continuous assessment of cerebral autoregulatory status might allow for more precise MAP and CPP titration and whether this could lead to avoidance of overly aggressive therapeutic interventions. They concluded that targeting the CPPopt six times daily using a bedside research software was both safe and feasible, encouraging a larger phase 3 trial. This will potentially elucidate whether this patient-specific, autoregulation-based therapy approach to precision medicine will improve outcomes in severe TBI patients.

In addition to standard ICP monitoring, many centers incorporate the use of brain tissue oxygenation (Pbt02) monitoring. The general premise behind invasive ICP monitoring is to maintain adequate oxygen delivery to the brain. CPP is typically used as an estimate of cerebral blood flow. However, factors other than blood flow may limit tissue oxygenation, most notably cerebral edema which can impact oxygen diffusion. It was previously shown that brain hypoxia was independently associated with poor prognosis and poor short-term outcome in severe TBI [71]. In a phase II trial (BOOST-II), patients treated in the Pbt02-directed protocol had reduced brain tissue hypoxia compared with those treated in the ICP-only protocol leading to a trend toward improved functional outcomes at 6 months [72]. Further evaluation is ongoing in the phase 3 BOOST-3 trial to determine if a treatment protocol based on Pbt02 plus ICP monitoring improves the GOS-E at 6 months compared with ICP monitoring alone [73].

### 3.4. Surgical Management

Two randomized trials regarding the role of decompressive craniectomy in severe TBI patients were found to have unfavorable (vegetative state) long-term outcomes [9,10]. However, the timing of the craniectomy was different; the DECRA trial had the craniectomy performed in the first-tier management while RESCUEicp trial enrolled the patient as last tier intervention. The mortality in late-stage intervention resulted in decrease in mortality and similar long-term outcomes. The debate continues regarding the role of the decompressive craniectomy in severe TBI. Future study may identify a cohort of patients that will receive benefits with the decompressive craniectomy.

The most common indication for surgical intervention in TBI is the evacuation of an acute subdural hematoma. The role of craniotomy vs. craniectomy in managing acute traumatic subdural hematoma was the subject of the RESCUE-ASDH trial. In this multicenter randomized trial, patients with an acute subdural hematoma that warranted evacuation via a large bone flap were randomized intraoperatively to either have their bone replaced (craniotomy) or to have their bone left out (craniectomy). The primary outcome was the GOS-E at 12 months. Primary outcomes were similar between groups. Patients in the craniotomy group required additional surgeries more frequently than the craniectomy group, but the craniectomy group had more wound-related complications [74].

#### Cisternostomy

DC has been shown to successfully reduce ICP and decrease mortality in patients with TBI. However, neither craniectomy nor ventricular drainage directly reduce cerebral edema. It has been proposed that performance of a cisternostomy with or without cisternal drain placement can improve cerebral edema via the glymphatic system. Traumatic subarachnoid hemorrhage in TBI can block the normal CSF pathways leading to a rise in cisternal pressure, decreasing interstitial fluid drainage, causing cerebral edema. Opening the basal cisterns to atmospheric pressure may allow interstitial fluid to drain out of the brain parenchyma thereby reducing cerebral edema. In a single-center RCT comparing DC alone vs. DC plus basal cisternostomy without cisternal drain placement, it was found that the addition of the basal cisternostomy effectively reduced ICP, duration of ventilation, ICU length of stay, and mortality and resulted in a better GOS-E at 12 weeks [75].

In another study, investigators randomized patients to undergo standard DC versus primary cisternostomy to evaluate whether cisternostomy can replace decompressive craniectomy in TBI. They found that cisternostomy reduced ICP, days on ventilatory support, and ICU length of stay as well as mortality when compared to craniectomy. They proposed that cisternostomy allows for avoidance of a second operation which is required to replace the bone flap in the case of craniectomy but were unable to claim cisternostomy as a replacement for craniectomy due to the small sample size [76]. Large randomized controlled studies are needed to evaluate the role of cisternostomy in the management of severe TBI.

### 3.5. Limitations

Our review was performed from the published report; however, some recommendations were from level II or below. Well-designed studies are needed to validate some of the current management recommendations and explore the new modalities in the management of severe TBI.

## 4. Conclusions

BTF guidelines provide armamentarium of management of severe TBI. The goal of medical management including maintenance of certain parameters of blood pressure, ICP, CPP, and oxygenation and ventilation is to minimize the secondary insult of severe TBI. Antiseizure prophylaxis, proper sedation, and appropriate paralytic use also help mitigate further deterioration of the neurological condition. DC can be performed to reduce ICP, however, the timing of DC is still debatable, and the long-term neurological outcomes have not been consistent. Ongoing research provides some evidence of beneficial effect with new therapeutic intervention, diagnostic and prognostic biomarkers, and new operative approaches to manage TBI. Further high-quality research is needed to explore the new modalities and validate the effectiveness of the modalities.

## Data Availability

Data sharing is not applicable.

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
