# Peer review of "Current Management and Future Challenges in the Management of Severe Traumatic Brain Injury"

_medicina, 2025, doi:10.3390/medicina61040738_

Round 1
Reviewer 1 Report
Comments and Suggestions for Authors
Abstract:
Although the background is given in the abstract, the main idea that is intended to be explained in the material, methods, and results is not clear. Please indicate what you want to focus on in your study and what parameters you evaluate while doing this.
Introduction:
The introduction does not provide sufficient information about the main topic of your manuscript. There is a gap between paragraphs. Please ensure the integrity of the flow. Make brief remarks about what awaits the reader in the content. The hypothesis of the review was not mentioned, and the author did not talk about the study design or why they wanted to review the TBI.
The information given about decompressive craniectomy does not state why it is examined in two categories and which references were used to select these examinations. Please make additions.
In the study examples given for decompressive craniectomy, the characteristics of the patients were not mentioned. In addition, the hours at which the results were obtained differed. Do these have any significance for treatment? Please explain in more detail.
The mechanism of Hyperosmolar Therapy is not written very clearly. Expand the treatment mechanism and give more recent literature examples.
Babiturates have been said to inhibit brain metabolism, but no explanation has been given as to how they do this. In order to understand the effects of the drugs, their mechanisms of action should be briefly discussed.
Conclusion:
Thank you for providing an instructive study, but mentioning the limitations of the study may help authors who wish to undertake similar studies gain insight.
Comments on the Quality of English LanguageThere are spelling errors in your manuscript. Please check it (e.g., "that it is crucial to maintain hemodynamic stability prior to and during barbiturate" etc.)
Author Response
Manuscript ID: medicina-3556106
Open Review
( ) I would not like to sign my review report
(x) I would like to sign my review report
Quality of English Language
(x) The English could be improved to more clearly express the research.
( ) The English is fine and does not require any improvement.
Is the work a significant contribution to the field?
Is the work well organized and comprehensively described?
Is the work scientifically sound and not misleading?
Are there appropriate and adequate references to related and previous work?
Comments and Suggestions for Authors
Abstract:
Although the background is given in the abstract, the main idea that is intended to be explained in the material, methods, and results is not clear. Please indicate what you want to focus on in your study and what parameters you evaluate while doing this.
Answer. first, I would like to thank the reviewer for taking time to review our manuscript and provided with very valuable feedback. As suggested, we have clarify the methods and results section of the abstract
Introduction:
The introduction does not provide sufficient information about the main topic of your manuscript. There is a gap between paragraphs. Please ensure the integrity of the flow. Make brief remarks about what awaits the reader in the content. The hypothesis of the review was not mentioned, and the author did not talk about the study design or why they wanted to review the TBI.
Answer. Introduction was revised based on suggestion.
The information given about decompressive craniectomy does not state why it is examined in two categories and which references were used to select these examinations. Please make additions.
Answer. we have removed the detailed of the decompressive craniectomy part from the future challenges category. We just kept the small paragraph to show that there is still a debate among the neurosurgical community regarding the procedure.
In the study examples given for decompressive craniectomy, the characteristics of the patients were not mentioned. In addition, the hours at which the results were obtained differed. Do these have any significance for treatment? Please explain in more detail.
Answer. we have expanded the patients’ characteristics in the decompressive craniectomy section.
The mechanism of Hyperosmolar Therapy is not written very clearly. Expand the treatment mechanism and give more recent literature examples.
Answer. Mechanism of the action of the hyperosmolar therapy was clarified.
Babiturates have been said to inhibit brain metabolism, but no explanation has been given as to how they do this. In order to understand the effects of the drugs, their mechanisms of action should be briefly discussed.
Answer. Barbiturates mechanism has been added to the section.
Conclusion:
Thank you for providing an instructive study, but mentioning the limitations of the study may help authors who wish to undertake similar studies gain insight.
Comments on the Quality of English Language
There are spelling errors in your manuscript. Please check it (e.g., "that it is crucial to maintain hemodynamic stability prior to and during barbiturate" etc.)
Answer. we reread that the manuscript for any typo grammatical errors.
Submission Date
13 March 2025
Date of this review
28 Mar 2025 18:25:26
© 1996-2025 MDPI (Basel, Switzerland) unless otherwise stated
Reviewer 2 Report
Comments and Suggestions for Authors
Thank you very much for letting me review this very interesting article “Current management and future challenges in the management of severe traumatic brain injury”.
This well-structured and engaging review provides a timely update on the current recommendations for management of severe traumatic brain injury. This review is a valuable resource, distilling key insights from years of research and providing a comprehensive outlook that encompasses current management approaches, forthcoming challenges, and innovative therapeutic strategies presently under investigation in clinical trials. It serves as a comprehensive reference guide, outlining the BTF guidelines and recommendations in the management of severe TBI.
To further strengthen the review, I propose that the authors consider incorporating updates on following points:
1) Prehospital airway management.
2) Post-TBI rehabilitation strategies.
3) Therapeutic potential of amantadine, ONP-002, Cerybrolysin.
Author Response
Manuscript ID: medicina-3556106
Open Review
( ) I would not like to sign my review report
(x) I would like to sign my review report
Quality of English Language
( ) The English could be improved to more clearly express the research.
(x) The English is fine and does not require any improvement.
Is the work a significant contribution to the field?
Is the work well organized and comprehensively described?
Is the work scientifically sound and not misleading?
Are there appropriate and adequate references to related and previous work?
Comments and Suggestions for Authors
Thank you very much for letting me review this very interesting article “Current management and future challenges in the management of severe traumatic brain injury”.
This well-structured and engaging review provides a timely update on the current recommendations for management of severe traumatic brain injury. This review is a valuable resource, distilling key insights from years of research and providing a comprehensive outlook that encompasses current management approaches, forthcoming challenges, and innovative therapeutic strategies presently under investigation in clinical trials. It serves as a comprehensive reference guide, outlining the BTF guidelines and recommendations in the management of severe TBI.
Answer. we thank the reviewer for taking time to review the manuscript and provided with very valuable comments.
To further strengthen the review, I propose that the authors consider incorporating updates on following points:
1) Prehospital airway management.
Answer. we have added a paragraph regarding the prehospital airway management.
2) Post-TBI rehabilitation strategies.
Answer. Post-TBI rehabilitation strategies are incorporate in the manuscript.
3) Therapeutic potential of amantadine, ONP-002, Cerybrolysin.
Answer. We have added a paragraph of some experimental therapy into the manuscript.
Submission Date
13 March 2025
Date of this review
01 Apr 2025 00:53:50
Round 2
Reviewer 1 Report
Comments and Suggestions for Authors
Thank you for the revisions.
Academic Editor Notes
Please amend paragraphs 2 and 3 of the Introduction for clarity, The additions suggested by the reviewers were important however there is now some flow issues and duplication across these two paragraphs. Some specific suggestions:
Answer. Thank you for the valuable feedback.
Line 43-45 - Please provide some additional detail on who utilises the guidelines produced by the BTF (e.g., USA or global) and whether these would be supplementary to the the guidelines produced by ACS.
Answer. we provided additional detail in the paragraph. We don’t think we need to link the guidelines to our manuscript. We have provided the reference if some one like to access those recommendations.
Line 45 - Please provide more detail regarding how evidence-based care in severe TBI resulted in a reduction in mortality and morbidities (given the high incidence of mortality that remains).
Answer. we provided more detail in the line.
Line 46-47 - Who published the most recent update in managing severe TBI (was this BTF?)
Answer. Elaborated further about the recent updates.
Line 47 - The statement that severe TBI patients are the most challenging patients to manage would be better placed at the start of the paragraph (i.e. as an introduction to the problem).
Answer. As suggested, the statement was placed at the beginning of the paragraph. I must agree that it flows much better. Thank you.
Line 47-49 - Some of this detail is now replicated in the added paragraph. Ideally, the information about the review aim and inclusions should be left for the final introduction paragraph, with the second paragraph focused on the problem, current knowledge and the gap in the literature that exists which justifies this review (i.e., Line 49-53).
Paragraph 3 - Start with a clear aim, i.e., "The aim of this review is to present current management practices and future challenges in severe TBI care". This can be followed by Lines 55-59 (i.e. hypothesis, review summary).
Answer. As suggested 3rd paragraph was adjusted based on the suggestion.
Line 52 - The statement that current management evolved over the period needs clarity (i.e., over what period - since guidelines were first produced?).
Answer. statement was added to clarify the period.
Categorization and order of the current management strategies could be improved to be more logical, e.g.:
2.1 Surgical Management - Decompressive craniotomy
2.2 Pharmaceutical Management
2.2.1 Hyperosmolar Therapy
2.2.2 Seizure Prophylaxis
2.2.3 Sedatives
2.2.4 Barbiturates
2.2.5 DVT Prophylaxis
2.3 Medical Management
2.3.1 Prehospital Airway Management
2.3.2 Ventilation
2.3.3 Cerebrospinal Fluid Drainage
2.3.4 Nutrition
2.4 Patient Monitoring
2.4.1 ICP Monitoring and Thresholds
2.4.2 Blood Pressure Thresholds
2.5 Post-TBI Rehabilitation Strategies
Note that if Amantadine and Cerebrolysin are not currently recommended in any guidelines due to insufficient evidence (given the specific aim of your review) these would be better placed under 3.1 Therapeutic Strategies together with TXA and Atorvastatin.
Answer. Manuscript has been rearranged as suggested above.